# Insights into Batch Selection for Event-Camera Motion Estimation

**DOI:** 10.3390/s23073699

**Published:** 2023-04-03

**Authors:** Juan L. Valerdi, Chiara Bartolozzi, Arren Glover

**Affiliations:** Event-Driven Perception for Robotics, Istituto Italiano di Tecnologia, 16163 Genova, Italychiara.bartolozzi@iit.it (C.B.)

**Keywords:** dynamic vision sensor, pose estimation, neural network, deep learning

## Abstract

Event cameras measure scene changes with high temporal resolutions, making them well-suited for visual motion estimation. The activation of pixels results in an asynchronous stream of digital data (events), which rolls continuously over time without the discrete temporal boundaries typical of frame-based cameras (where a data packet or frame is emitted at a fixed temporal rate). As such, it is not trivial to define *a priori* how to group/accumulate events in a way that is sufficient for computation. The suitable number of events can greatly vary for different environments, motion patterns, and tasks. In this paper, we use neural networks for rotational motion estimation as a scenario to investigate the appropriate selection of event batches to populate input tensors. Our results show that batch selection has a large impact on the results: training should be performed on a wide variety of different batches, regardless of the batch selection method; a simple fixed-time window is a good choice for inference with respect to fixed-count batches, and it also demonstrates comparable performance to more complex methods. Our initial hypothesis that a minimal amount of events is required to estimate motion (as in contrast maximization) is not valid when estimating motion with a neural network.

## 1. Introduction

Event cameras [1] transmit a digital event at a pixel location when the pixel’s relative brightness change surpasses a threshold. Compared with standard "frame-based" cameras, event cameras offer several advantages, including low latency, high dynamic range, high temporal resolution, and signal compression. They inherently detect changes, making them particularly suited for estimating motion and potentially game changers in a variety of tasks, including visual odometry [2,3,4], simultaneous localization and mapping [5], structure from motion, object tracking [6], as well as monitoring and security [7]. The sub-millisecond temporal resolution makes them potentially extremely accurate for these tasks; whether motion comes from an externally moving object, or the camera itself.

Traditional approaches to estimating visual changes, such as by Lucas-Kanade [8], assume pixel-intensity consistency over short time periods, i.e., between two sequential images. Problems with “motion-blur” (movement during photon integration) must be overcome to produce viable results [9]. In contrast, event cameras produce asynchronous events at microsecond temporal resolution. The precise timing allows “motion-blur” to become separable and becomes an informative signal that can be used to calculate motion.

Camera motion estimations from events have been performed using iterative filters [5,10,11], optimization techniques [12,13,14,15], or trained neural networks [16,17]. While the techniques differ, they must all extract the same information from the same data flow, as shown in Figure 1. Therefore, they all compute some form of spatiotemporal gradients as the scene texture passes across multiple pixels during motion.

To have such spatiotemporal patterns available for extracting motion patterns, batches of more than one event must be processed. Figure 1 shows the spatiotemporal patterns in an event batch and the “motion-blur” that occurs if integrated into a single time step. While optimization algorithms operate directly on the list of <x,y,t> data, neural network implementations require these data transformed into a 3D tensor to be fed to the network. Event iterative filters, which can operate event-by-event, can only converge after processing more than one event and, therefore, have used batches as well [14,18].

As event cameras do not have a fixed frame rate, the batch of data to be considered simultaneously is not determined by the sensor. Instead, the size of a batch can be determined by algorithms, e.g., based on a short (millisecond) or long (seconds) period of time, and this paper investigates the optimal selection of data for a motion estimation task. A batch that is too large can violate the assumption of linear motion over small time periods that many algorithms require, such as [12,13,14,18], while at least some minimal amounts of data are required to form the necessary spatiotemporal gradients. A common hypothesis is that an optimal balance of information for camera rotation estimation could be achieved with 2 to 3 pixels of spatial motion.

The problem of batch selection, which refers to the *number* of events used for computation, should be decoupled from the complementary problem of batch encoding, which is *how* the tensor is formed from the data (and also an open question [16,17]). Batch selection for neural network-based algorithms has not been addressed in much detail and we aim to address the question of optimal batch selection for training and inference of neural networks for rotation estimation.

We utilized a current state-of-the-art network designed for camera pose change estimation [16] and modify it for rotation-only motion. We proceed to retrain the network using various batch-selection methods, including common techniques in event-driven vision, such as fixed-time windows and fixed event counts. Additionally, we introduce alternative methods that we devised, such as a `fixed rotation’ window, a network to estimate optimal batch sizes, a recursive strategy that determines the batch size based on the previously estimated result, and an ad hoc method for estimating batches of sufficient motion (referred to as *local*). We conduct experiments to evaluate the performance of each batching technique for both model training and inference.

Although camera rotation estimation is an important topic in its own right [13,19], we used a currently established method and investigated how the batch selection affects performance. We specifically chose a 3-DoF task to remove complexities that may introduce uncorrelated errors (e.g., from 6-DoF and depth estimation). The main contribution of this work is the insight into the optimal batch-selection technique for motion estimation using event cameras.

### 1.1. Related Work

Camera rotation estimation is the task of approximating the movement of the camera in 3-DoF (roll, pitch, yaw) given the visual camera input. Such a problem should be considered within the wider field of visual SLAM [19], which includes both camera pose estimation and map creation. For a mono-camera with rotation-only motion, no parallax occurs and depth cannot be estimated, leaving only a spherical panoramic map [20].

On the task of pure visual odometry [21], estimating the rotation change ΔR over a small time interval does not necessarily require a global map, and can be solved using epipolar geometry and key-point correspondence [22]. Incorrect calibration, model estimation, and data association can lead to errors in the estimated pose change, resulting in a drift in the integrated signal. To mitigate such problems, deep learning techniques have recently shown improved performance over geometrical techniques [2,3,4], at the expense of requiring large amounts of training data. Visual changes have been estimated in event-driven deep networks by providing image sequences [2], by explicitly storing the state used for comparison to individual images [3], or by inherently storing the state in a recursive neural network [4].

RGB images provide a measure of the light intensity at each pixel at a fixed point in time. Event cameras produce different visual signals, asynchronously measuring visual changes. When a change is detected, they return an event e=(x,y,t,p). The components (x,y) represent the position of the pixel that detects the change, *t* is the timestamp, and is *p* the polarity (brighter or darker change). An individual event at a single pixel location does not contain enough information on its own; therefore, multiple consecutive and neighboring events must be processed to understand the scene and motion.

Geometric approaches to visual odometry with event cameras are more similar to `direct visual odometry’ [23], as features are not extracted; instead, the pose change is computed directly over the raw data by computing the spatiotemporal warping required to maximize the contrast of a batch of events [13,15,24,25,26,27]. The estimation of this vector is achieved through optimization and the batch size needs to be large enough for the convergence of the optimization, but small enough to avoid the superposition of complex motion that cannot be described with a single warping. Learning-based approaches for the estimation of optical flow, depth, and camera motion from event cameras use three channel images (representing time and event polarity counts) [17] or an <x,y,t> normalized 3D tensor [16] as input to deep convolutional neural networks. The encoding of events into the tensor has an impact on the performance; however, *which* events are selected (the focus of this work) is given less attention. Specifically, the choices of fixed-time events and fixed-number batches of events are often made without clear motivation, and other selection methods have not been thoroughly explored.

An event batch of fixed-time length contains a highly variable number of events, which depends on the camera and independent object motion. As such, the assumption of linear motion cannot be guaranteed for long temporal windows or large speed motions. Conversely, for short temporal windows or small motions, the information in the batch can be too small. An event batch with a fixed number of events may guarantee a minimum amount of information to be processed, but it can also vary depending on the scene texture. Therefore, other methods to decide *which* events should be used in a single tensor for processing could improve the overall network performance.

## 2. Materials and Methods

The focus of this work is to investigate batch selection methods for motion estimation using learning-based approaches. We used a state-of-the-art network to perform motion estimation and simplify the problem to rotation-only to isolate errors due to inaccuracies in scene depth, as well as those due to the entangling of rotation and translation motion.

### 2.1. Rotation Estimation Network

We simplified EV-FlowNet v2 [16] by removing the depth decoder and using only the pose model related to rotations, as shown in Figure 2.

As input, the network uses a tensor that attempts to maintain the spatiotemporal information of the events by discretizing the time domain into a set of bins. Given a selection of *N* input events $\{\left(x_{i},y_{i},t_{i},p_{i}\right),i=1,2,\ldots ,N$}, and a set of *B* temporal bins, we define the input tensor V(x,y,t), where (x,y) are the image coordinates and t=0,1,…,B−1, as follows:$$\begin{array}{ccc} V(x,y,t) & = & \sum_{i}p_{i}k_{b}\left(x-x_{i}\right)k_{b}\left(y-y_{i}\right)k_{b}\left(t-t_{i}^{*}\right)\\ t_{i}^{*} & = & \left(B-1\right)\left(t_{i}-t_{1}\right)/\left(t_{N}-t_{1}\right)\\ k_{b}\left(a\right) & = & \max(0,1-|a|) \end{array}$$

The output of the network is the predicted change in the camera rotation pose, ΔX, between the time points $t_{1}$ and $t_{N}$.

Training is performed using a supervised approach instead of self-supervised, as in [16], with the goal of increasing the likelihood of producing a well-trained network and isolating performance differences based on the batch-selection method, rather than other sources. We created new simulated datasets with diverse characteristics to evaluate the network and batch selection methods in a variety of scenarios. Supervised learning on simulated data has been used successfully in recent work [28,29,30].

### 2.2. Batch Selection

The input to the rotation estimation network, for both training and inference, is a fixed-size 3D tensor that can be filled with a variable number of input events or event batches. Converting an event batch to a tensor is described in Section 2.1, while the criteria to select event batches is the major focus of this work. Every event batch covers a discrete:1.Number of events;2.Temporal period; and3.Camera rotation.
Any of these can be used to define the criteria to select events from the event stream. We trained a rotation estimation network for each batch method.

#### 2.2.1. Fixed Number of Events (Win*T*)

The batch is always formed from a fixed and constant number of events *N*, as in [12,15,16,18,26,27,31,32]. A fixed count of events ensures that there are always some events available to infer motion. However, for low-texture scenes, the batch may not be filled until a large (possibly non-linear) motion is performed.

#### 2.2.2. Fixed Temporal Period (*N*K)

The batch is formed from all events in a set time period, *T*, as in [14,29,33,34,35,36]. A temporal window has data proportional to the speed of the camera and, therefore, results in a wide variance in the number of motions encoded in the tensor.

#### 2.2.3. Fixed Camera Rotation: Local

As texture varies across different scenes, a fixed camera motion will produce batches of different sizes. However, at the scale of a single object, a stronger correlation between camera velocity and the number of events exists. The *local estimation* method uses a measure of the percentage of events that have occurred in small neighborhoods of pixels (at the scale of a single object edge), with the idea that a moving edge will activate all pixels in the small region after a fixed, repeatable, amount of motion. For each event, the local method evaluates multiple adjacent patches to estimate the percentage of active events.

Given the set of events since the last batch, $E=\{\left(x_{i},y_{i},t_{i},p_{i}\right),\;i=1,\ldots ,N\}$, we define the image of *n* events, n≤N, as
$$I_{n}\left(x,y\right)=1,$$
if there exists $e_{i}\in E,i\leq n$, such that $\left(x,y\right)=\left(x_{i},y_{i}\right)$.

For each event $e_{n}$ that has occurred since the last batch was formed, we check four adjacent m×m patches. Each patch is adjacent to the event along the x-axis or y-axis; in Figure 3, we show an example. A batch of *n* events is formed if the event $e_{n}$ has at least one adjacent patch in which:(1)$$\frac{\sum_{j=-m}^{m}\sum_{k=-m}^{m}I_{n}\left(j,k\right)}{m^{2}}>T_{le}$$
where the *local estimation* threshold, $T_{le}$, is approximately 0.13.

#### 2.2.4. Fixed Camera Rotation: Recursive

The recursive estimation selects a batch of events with a predefined magnitude of motion, assuming that the previous velocity estimate is similar. Given the previously predicted pose change estimate ΔX, the rotational speed *v*, is calculated as:$$v_{i-1}=\frac{\left|\Delta X\right|}{t_{i-1}}$$
and, therefore, the time window to produce a similar amount of motion for $t_{i}$ is:$$t_{i}=\frac{r}{v_{i-1}}+k$$
where *r* is the desired rotational change and *k* is a small offset.

Such an approach assumes an accurate and stable estimation of ΔX. Any incorrect estimation can lead to divergence of the algorithm over time. In practice, depending on the algorithm, an offset value, such as *k*, prevents such a divergence.

#### 2.2.5. Fixed Camera Rotation: Trained Network (Net)

Another method to estimate an appropriate batch of events for a constant magnitude of rotation is to use a small trained network. The input to the network is a 3D tensor, *I*, with each slice $I_{i}$ being an image of accumulated events on the visual plane for a number of events, $N_{i}$:$$I_{i}\left(x,y\right)=1\;\mathrm{if}\exists e_{j}\left(x,y\right),\;j\in \left[0,N_{i}\right].$$
Example images of typical layers are shown in Figure 4b.

The output of the network is an estimate of the *N* events that best match the desired fixed rotation amount. The architecture of the neural network is shown in Figure 4a.

The design decision lies in selecting the value of $N_{i}$ for each layer, which affects the temporal resolution represented by the fixed-size tensor and sets the minimum and maximum velocity that can be estimated. The amount of texture in the scene is the main factor that influences the number of events occurring during any fixed rotation. More structure, edges, and objects produce a greater number of events, given the same camera motion. As the field of view is two-dimensional, the number of events also scales exponentially, and must be accounted for in the selection of *N*:(2)$$N_{i}=a\beta^{i}$$
where *a* is the smallest possible event batch and β>1 is the rate scaling factor. Therefore, the range of possible values of *N* falls in the interval $\left[a,a\beta^{n}\right]$ for $N_{i},1\leq i\leq n$.

#### 2.2.6. Fixed Camera Rotation: Ground-Truth

The ground-truth camera rotation can be used to form batches of events *for training only*. Therefore, we compare the above estimation methods against networks trained with perfect knowledge of camera rotations. These networks must use one of the above estimation methods for inference.

External sensors, such as inertial measurement units, could be used to estimate a constant rotational motion, and then fused with visual information. In this paper, we focus on vision-only solutions for scenarios in which other sensors are not available.

## 3. Results

We evaluate the network performance at estimating camera rotations for the different batch methods used in training. The performance is measured as the absolute difference between the estimated rotation velocity and the ground truth, over each trajectory.

### 3.1. Datasets

Simulated datasets were used to provide accurate ground-truth for camera motion and to ensure that it was restricted to pure rotations. We selected four publicly available photorealistic simulated environments from the UnrealCV Zoo [37], which included diverse lighting, textures, shadows, reflections, and object clutter, as shown in Figure 5. The camera was positioned inside the virtual room and randomly rotated along all three axes at a variety of speeds, as shown in Figure 5e. Frames were generated at a rate of over 1 kHz from which the event stream was generated using log-image-difference techniques [38]. We used five trajectories for *arch1*, four trajectories for *arch2*, and three trajectories for *arch3* and the final *room* dataset, resulting in a total of 13 different datasets (each with a different velocity trajectory) covering a total of 190 seconds of data.

### 3.2. Training

Datasets were split into 150 s of training (all *arch* datasets) and 40 s of testing (the single previously unseen *room* dataset), which were spread across each of the datasets in Figure 5e. Event batches were created in pre-processing using each batching method at predefined fixed intervals in the dataset (every 5 ms) to ensure a valid comparison. Each batch was then converted to the required tensor format, as described in Section 1.1.

The motion estimation networks and the selection network were trained using the Adam optimizer with learning rates ranging from 0.005 to 0.0001 and a batch size of 256. The motion networks were trained for 1000 epochs using an Nvidia Quadro RTX 6000, which took approximately 10 h. The selection network was trained for 200 epochs, which took approximately 1 h.

The selection network parameters, as defined in Equation (Equation 2), were set as follows: a=4000, β=1.34, and n=16. These parameters define a predictability range of [4000, 432,251] events. To ensure that the network learns equally at different scales within this range, we used the mean squared logarithmic error (MSLE) as the loss function.

### 3.3. Batch Selection Parameters

Batch parameters were chosen to ensure linear motion within each batch [12,13], resulting in valid EV-FlowNet v2 performance. Figure 5f shows that the rotation parameters of $0.5^{\circ }$ and $1.0^{\circ }$ lead to an edge blur of approximately 1–2 pixels and 3–5 pixels, respectively. As a control, we also trained a network using batches with random rotations between 0.5 and 1.0 degrees.

The parameters selected for each trained model are summarized in Table 1 and an example of the resulting batch size is shown in Figure 6. Fixed number batches were chosen as 30K, following [12,16,24,27], and 100K after observing typical amounts of texture in our datasets. The fixed-time batch of 20 ms resulted in a mean of $∼0.57^{\circ }$ of the rotation and 50 ms resulted in a mean of $∼1^{\circ }$ of the rotation. The selection network was trained with a target of $1^{\circ }$. The local estimation method was heuristically tuned from trial and error, as noise and artifacts influenced the resulting batches in unpredictable ways.

Fixed rotation methods can only be used for network training under operational conditions as they require ground-truth information. In Table 1, we label them as *train-only*. However, for baseline purposes, we also perform inference with fixed rotation batches since we have ground-truth available in our experiments. The recursive methods are labeled *infer-only*.

### 3.4. Batch Size Analysis

Fixed-time batches vary with velocity and scene texture but follow a similar trend to fixed rotation batches, which only vary with scene texture. The selection network (Net) and the recursive strategy (Rot1.0${}_{r}$) were designed to mimic Rot1.0 batches. Both strategies performed similarly, with a logarithmic error of 0.23 for Rot1.0${}_{r}$ and 0.19 for Net. As shown in Figure 6b, they approximate the order of magnitude, peaks, and troughs of Rot1.0 well.

The local method did not follow Rot1.0 batch size in terms of the order of magnitude, and was unable to predict trends in the data, even though the parameters were tuned to achieve a similar amount of “edge-blur”. Figure 7 shows that the main reason for this behavior is that the activated pixel region meets the threshold due to texture from multiple nearby edges, rather than a single clear edge moving through the 7×7 region. This problem was consistently observed across different kernel sizes.

### 3.5. Camera Rotation Estimation

Multiple models were trained to estimate camera rotation, each using a different batch selection method. Inference was performed firstly using the same batch selection method used during model training, and secondly using all valid batch selection methods for each trained model. The performance was measured as the mean difference between the estimated velocity and the ground-truth velocity.

#### 3.5.1. Identical Batch Method for Training and Inference

The 100K model evaluated with the 100K batch is the poorest estimator compared to the other models, as shown in Figure 8. The batch size is significantly larger than the others and is likely not producing the required linear motion. The relative performance is visualized in Figure 9a.

The fixed rotation networks (Rot0.5, Rot1.0, RotR) generally have the lowest error and best performance, as shown in Figure 9b. However, Figure 8 shows that fixed-time networks (Win20, Win50) perform at a comparable level and outperform 30K and Net. As fixed rotations are not valid methods for live inference (and are presented here only for comparison to a baseline), the fixed-time batches actually give the best performance.

In summary, our results show that:Fixed-time networks vastly outperform fixed-count networks.The selection network (Net) did not match the performance of fixed rotations, indicating the batching methods are highly sensitive to the precision of the estimator.The local window shows the highest average error, which indicates that the algorithm itself did not achieve the desired result of measuring a consistent amount of rotation.

#### 3.5.2. Robustness of the Trained Models (Comparing Figure 10 Columns)

Comparing how different batch selection methods perform on models trained with other batch selection methods, as in Figure 10, provides insight into which models generalize well, and why the temporal window performs well. The networks trained with local windows and 100K batches did not perform well when tested with any of the batch selection methods, indicating that the model itself is not well-trained for the task.

The networks trained with RotR, Win20, and Win50 batches generalized well, as shown by the better performances during inference when using most of the other batch selection methods. Networks trained with fixed rotation batches produced very strong results when using the same batch selection method in the training (i.e., Rot1.0-Rot1.0 and Rot0.5-Rot0.5) but did not generalize as well during inference when using other batch selection methods.

The networks trained with 30K and Net batches produced consistent results but with a lower performance than the networks trained with the batch selection methods mentioned above.

#### 3.5.3. Suitability of Batch Methods for Inference (Comparing Figure 10 Rows)

Rot1.0, RotR, and Rot0.5 batch selection methods require motion ground-truth to produce the batches; therefore, they cannot be used during inference.

Fixed-time batches produce lower errors than fixed-count batches, across many of the models. The Net batches did not produce a well-trained model; however, it performed well as an inference method using models trained in other batches. Finally, the recursive method (which can be used for inference only), also had a high performance for the RotR model.

## 4. Discussion

### 4.1. Training with a Large Variety of Batch Sizes Is Important for Model Generalization

In retrospect, it is clear that a model trained on a wide variety of batch sizes and velocities can generalize well to any input batch. Therefore, the network becomes less dependent on a specific input batch method, and it can perform well with a range of batch selection methods. RotR produces the best-performing and most generalized model for many different inference batch methods, while Win20 and Win50 also produce well-generalized models. This has a promising implication that these networks can be trained with data augmentation by generating different batch sizes from the same data to increase the available training data multiplicatively.

### 4.2. A Minimum Amount of Rotation Is not Necessary for Deep-Learning Techniques

While contrast maximization techniques require multiple events from neighboring pixels warped onto a single pixel to enable optimization convergence, the same does not seem to apply to trained network approaches. Instead, the network is able to infer even very small amounts of motion from sparse event inputs. For example, it could be that the network uses the fact that *no neighboring events* occurred to estimate a small motion and the orientation of slight edges to estimate the direction of motion.

### 4.3. Inference Choice

The Net, recursive, and fixed-time batching methods all produced comparable inference results. However, Net is a secondary network that requires additional training data, which impacts run-time as it must be performed before each rotation estimation inference call. The recursive method can result in stability and convergence issues since large accelerations can invalidate the assumption that consecutive batches should have approximately the same size. If an unsuitable batch size results, the velocity is compromised for correctly predicting future batch sizes. Therefore, a simple fixed-time batching method is recommended for the inference phase.

Another surprising result is the performance difference between fixed-time and fixed-number batches. Both are simple metrics that, for a wide variety of event-driven algorithms, are sometimes arbitrarily chosen and alternated without justification. The results indicate that the choice can actually have a significant impact on the outcomes. For example, in [16], a fixed-number window is used, while even better results may be achieved if a fixed-time window was used instead.

In datasets with many directional changes, there are many points of zero motion (i.e., zero events), which cannot be represented by a fixed number of events. A window that includes motion from either side of the inflection point invalidates the assumption of linear motion. This could explain why the model trained with a batch size of 100K produced a error higher than 30K.

### 4.4. Local Windows

There are likely more effective methods than the one we proposed for detecting consistent motion through small local regions. However, since our results indicated that a fixed-time batch was sufficient, further investigation into local estimation methods was deemed unnecessary. This result provides further evidence that the actual spatiotemporal patterns of events that occur at a fine-grained resolution may not be as neatly organized as often assumed.

## 5. Conclusions

We evaluated various methods for selecting *which* events should be in a given batch that forms the input tensor to a camera rotation estimation network. We drew the following conclusions from the experiments:Network training should be performed with the widest variety of batch sizes and velocities to produce a well-generalized network.The fixed-time batch is recommended for inference; the exact size of which should depend on the application and typical velocity profiles of the camera. We believe this conclusion may serve as the baseline for any other task that measures velocity or optical flow, but may not be able to be extrapolated to tasks that measure absolute position, for example, object recognition.Networks for measuring pose change or velocity possibly have an advantage over contrast maximization techniques for measuring small velocities with a small batch.

The initial hypothesis that a minimum amount of events is required to accurately measure motion was found to be incorrect for the learning-based method used; therefore, the proposed methods for training a network, local motion estimation, and recursive strategies were not beneficial to the task.

The code and datasets are available as open-source to further contribute towards improving motion estimation with event cameras.

## Figures and Tables

**Figure 1 sensors-23-03699-f001:**
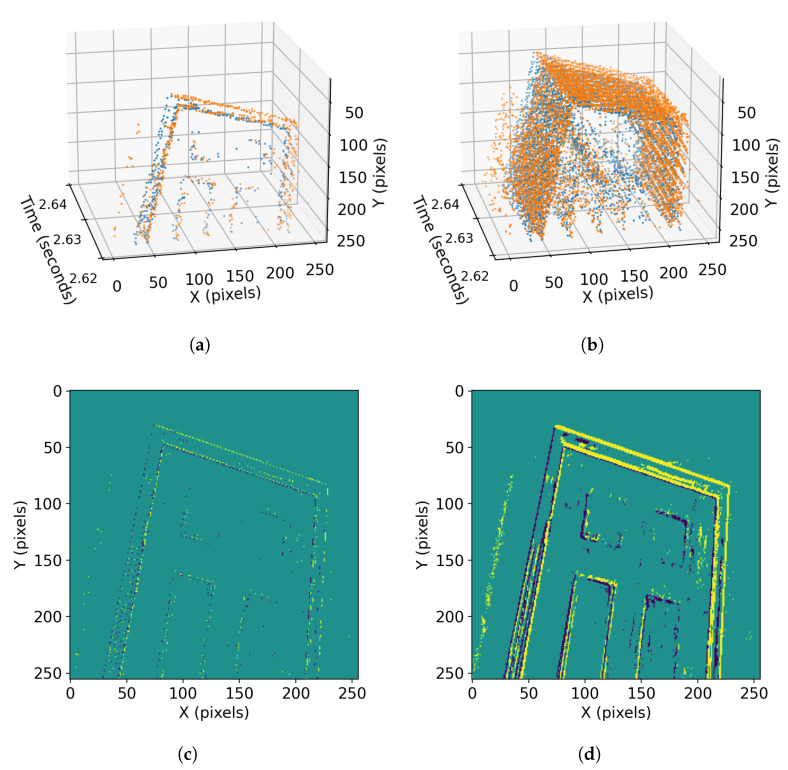
Batch selection for event-camera motion estimation, where colours indicate different polarity (increase or decrease in light) of events: (**a**) spatiotemporal volume of a batch of 1K events; (**b**) spatiotemporal volume of a batch of 10K events; (**c**) image formed with 1K events. It may provide understanding about the object (a door) but not the direction of the camera motion; (**d**) image formed with 10K events. It is a more appropriate number of events and can provide enough data to extract visual changes and, hence, camera motion.

**Figure 2 sensors-23-03699-f002:**
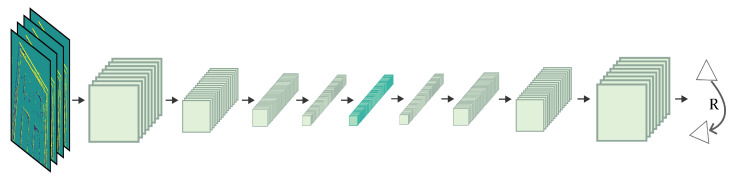
Network for camera rotation estimation, simplified from EV-FlowNet v2 [16].

**Figure 3 sensors-23-03699-f003:**
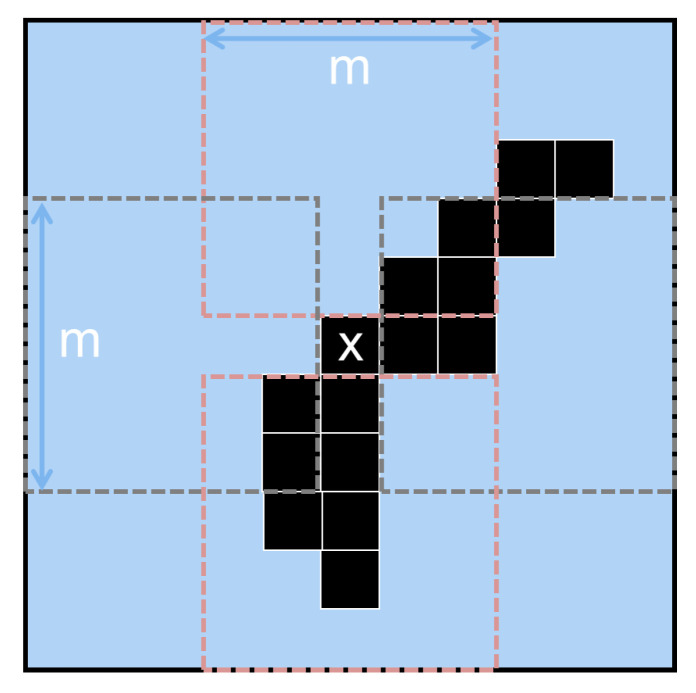
The local estimation method triggers a batch if any of the four regions of size *m* (red- and gray-dotted) surrounding the most recent event (center *X*) contains more than $m^{2}T_{le}$ pixels that have been triggered by previous events (black squares). In low-noise conditions, the criteria will be met when an edge passes through any of the dotted regions, accumulating events of more than one-pixel thickness.

**Figure 4 sensors-23-03699-f004:**
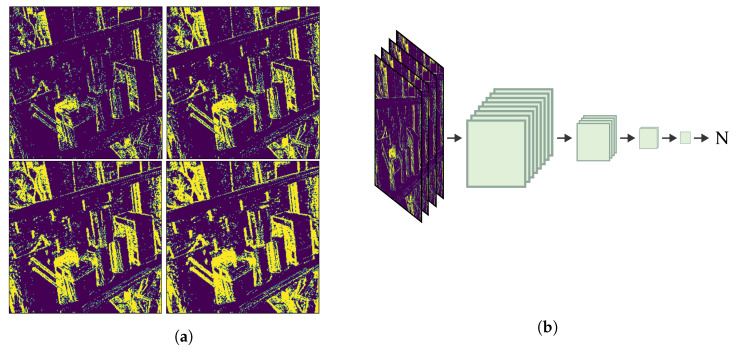
The selection network used to estimate the number of events *N* needed to produce a specific magnitude of rotation, without knowing the rotation itself. (**a**) Examples of input layers to the selection network showing $I_{25K}$, $I_{50K}$, $I_{75K}$, and $I_{100K}$, respectively, as the edges become bolder/thicker. (**b**) The selection network has four convolutional layers reducing the spatial–temporal information with each layer until obtaining a 1×1 tensor as output.

**Figure 5 sensors-23-03699-f005:**
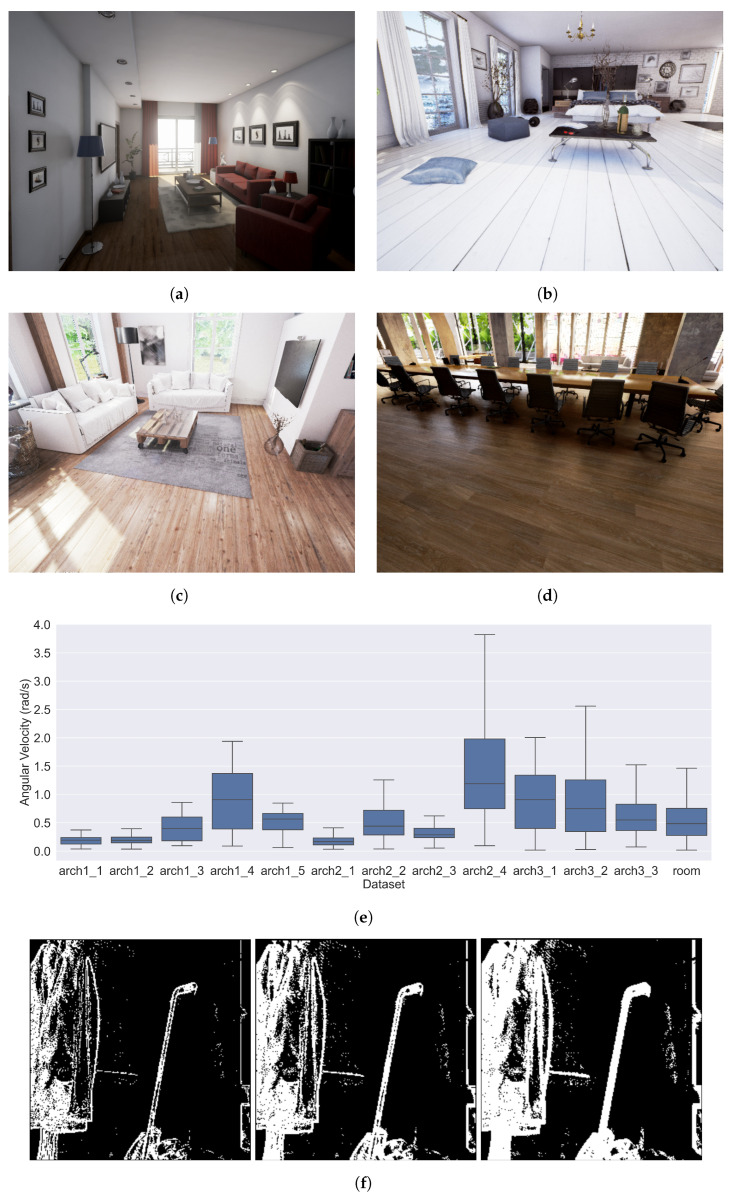
Example scenes (images from OpenCV website) (**a**) “room” from *RealisticRendering*, (**b**) “arch1” from *ArchinteriorsVol2Scene1*, (**c**) “arch2” from *ArchinteriorsVol2Scene2*, (**d**) “arch3” from *ArchinteriorsVol2Scene3*, and analysis of the simulated datasets: (**e**) angular velocity mean and variances of all simulated datasets, and (**f**) event images demonstrating rotations of $0.25^{\circ }$, $0.5^{\circ }$, and $1.0^{\circ }$, from left to right.

**Figure 6 sensors-23-03699-f006:**
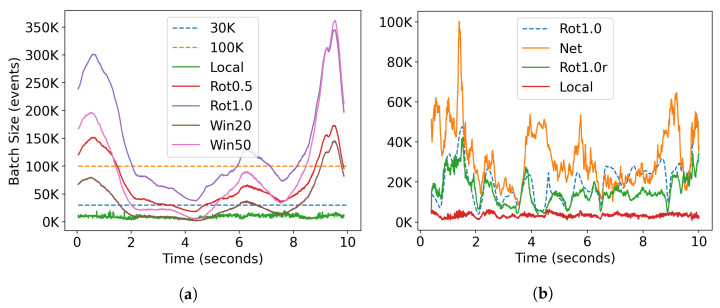
Batch sizes for different batch selection strategies: (**a**) arch1_1 dataset and (**b**) room dataset (first 10 s).

**Figure 7 sensors-23-03699-f007:**
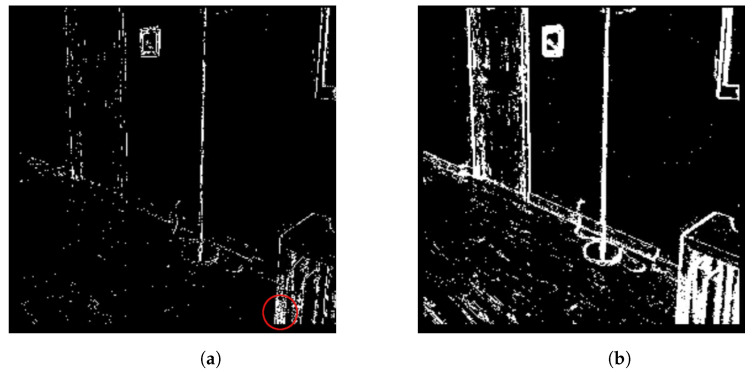
(**a**) Local vs. (**b**) Win20 event selection in the room dataset. Red circle represents the area where the local method conditions are met.

**Figure 8 sensors-23-03699-f008:**
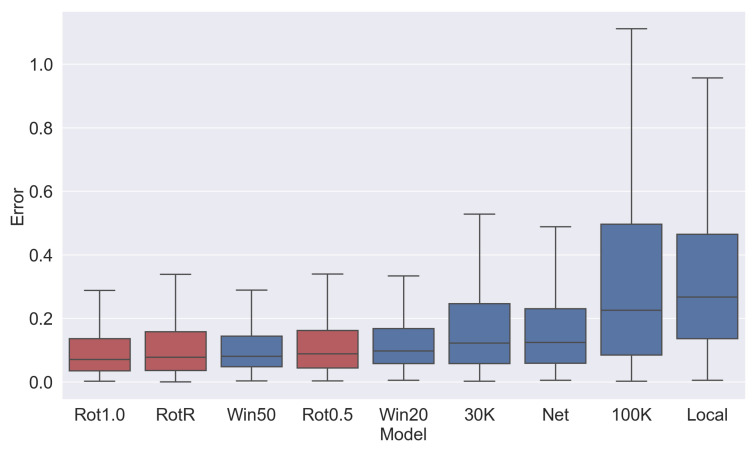
Rotation estimation error using the same selection strategy for training. Red bars indicate train-only results (cannot be used in a real system), while blue bars can be used for both training and inference.

**Figure 9 sensors-23-03699-f009:**
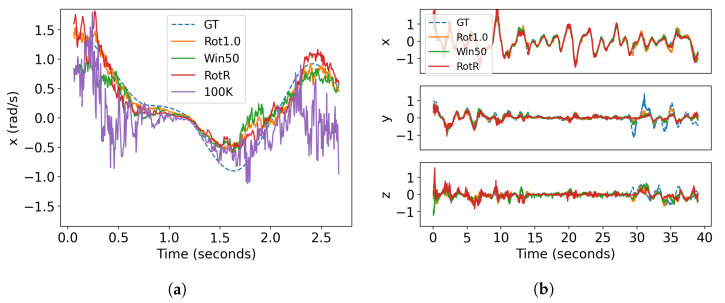
A comparison of estimated rotational velocities over time: (**a**) the first three seconds of x-velocity prediction on the room dataset and (**b**) the rotation velocity estimation compared to ground-truth.

**Figure 10 sensors-23-03699-f010:**
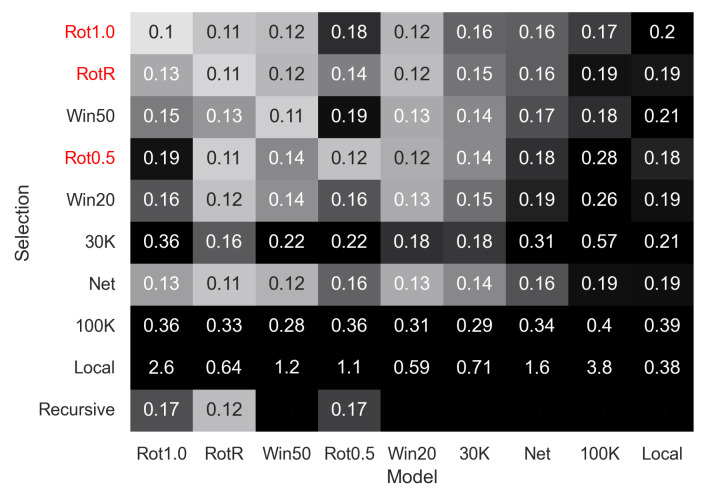
Error heatmap of all trained models evaluated with the different batch selection strategies. Selections written in red are train-only.

**Table 1 sensors-23-03699-t001:** Selection methods and motion estimation models used in the experiments.

Selection Method	Application	Parameter	Name
Fixed rotation	Train-only	$0.5^{\circ }$	Rot0.5
Fixed rotation	Train-only	$1.0^{\circ }$	Rot1.0
Random rotation	Train-only	$0.5^{\circ }$–$1.0^{\circ }$	RotR
Fixed count	Train/infer	30,000	30K
Fixed count	Train/infer	100,000	100K
Fixed time	Train/infer	20 ms	Win20
Fixed time	Train/infer	50 ms	Win50
Selection network	Train/infer	target $1.0^{\circ }$	Net
Local estimation	Train/infer	90%-7×7 patches	Local
Recursive	Infer-only	$1.0^{\circ }$ (trained Rot1.0)	Rot1.0${}_{r}$
Recursive	Infer-only	$0.75^{\circ }$ (trained RotR)	RotR${}_{r}$

## Data Availability

Datasets: https://zenodo.org/record/7762780, 10.5281/zenodo.77627 80 (last accessed: 20 March 2023); Code: https://github.com/event-driven-robotics/batch-selection-experiments (last accessed: 20 March 2023).

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
