# Peer review of "Insights into Batch Selection for Event-Camera Motion Estimation"

_sensors, 2023, doi:10.3390/s23073699_

Round 1

Reviewer 1 Report

This nice paper addresses data selection optimization for a motion estimation task from an event-based camera. The authors evaluated some selection methods considering the rotation estimation task. The work has scientific soundness and should be published, in my opinion.

Some comments:

In section 2.2. "Batch Selection", starting from line 151, the text is not clear enough on first reading. Authors should explain the various possible alternatives and make it clear which ones they will follow. This becomes clearer in the rest of the work, but should be clearly presented here.

Concerning to data sets, simulated datasets were used and I agree that it improves the  control of the experiment, but the authors must clarify the choice of scenarios and their characteristics. Do you tested other scenes??

Finally, the conclusion does not cover all the important results mentioned in the discussion. There should be greater detail and completeness.

The following statement is obvious and not necessary. “The article is presented to share our results with the scientific community…”

After minor revision, I think the paper should be published. It is very interesting.

Author Response

The authors appreciate and would like to thank the reviewer for the time and effort put in. We believe the manuscript improved based on the recommendations.

In section 2.2. "Batch Selection", starting from line 151, the text is not clear enough on first reading. Authors should explain the various possible alternatives and make it clear which ones they will follow. This becomes clearer in the rest of the work, but should be clearly presented here.

We have reformatted this section to clearly list the different types of batch selections we are using. We clarified in the text that we are comparing all of them. We have a a sub-section for each for the definitions to this end.

Concerning to data sets, simulated datasets were used and I agree that it improves the  control of the experiment, but the authors must clarify the choice of scenarios and their characteristics. Do you tested other scenes??

We have improved Section 3.1 to clearly define which datasets we used and justify the selection. We have stated that there are 4 different environments. For each environment we have different trajectories with different motion patterns. The environment and statistics are shown in Fig. 5e.

Finally, the conclusion does not cover all the important results mentioned in the discussion. There should be greater detail and completeness.

We have reviewed the discussion and conclusion sections and performed some minor changes, however, we believe that the lengthy discussion about the experiments results should stay in the discussion section. We have then summarised the conclusions of this discussion in the conclusion section succinctly. Some speculations and opinions are limited to the discussion section, as they cannot be fully drawn from the results presented.

The following statement is obvious and not necessary. “The article is presented to share our results with the scientific community…”

We have removed this sentence.

Reviewer 2 Report

The authors in this paper we consider neural networks for rotational motion estimation, and they investigate appropriate selection of event batches used to form input tensors.

General Comment

I enjoyed reading the manuscript. In my opinion the text is well structured and well written. Where possible, I suggest improving the quality of some figures, the English language and increase bibliographic references

It is generally not recommended to use the same words in the title and in the keywords: I suggest to the authors to change the same keywords also present in the title.

The grammar is sometimes tortuous: you may lose the reader's attention. It is great material, so why do yourselves the disservice of disguising this in the verbose language? Be succinct: there is a correlation between the high IF journals and high-citation papers and shorter length (everything, title, abstract, text). In this digital age, people do not have the time to navigate long texts with unwieldy language. Shorten and strengthen, for your own benefit.

Specific comments

I have no specific comments to make on the text.

Except for these minor changes, the article can be accepted in its current form for publication.

Author Response

The authors appreciate and would like to thank the reviewer for the time and effort put in. We believe the manuscript improved based on the recommendations.

Where possible, I suggest improving the quality of some figures, the English language and increase bibliographic references

We have reviewed and revised the entire manuscript making substantial edits where better phrasing and clarity could be used. We have revised each of the figures, and updated:

  • 1 - updated image and axis labels
  • 3 - create a completely new figure
  • 5 - axis labels and fontsize
  • 6 - axis labels and fontsize
  • 8 - axis labels and fontsize
  • 9 - axis labels and fontsize

We have reviewed the bibliographic references and added 3 more references. We believe the current references adequately cover the prior work in the field. Could the reviewer please be more specific if you believe more references are missing?

It is generally not recommended to use the same words in the title and in the keywords: I suggest to the authors to change the same keywords also present in the title.

We have revised the list of keywords to avoid overlap with the title.

The grammar is sometimes tortuous: you may lose the reader's attention. It is great material, so why do yourselves the disservice of disguising this in the verbose language? Be succinct: there is a correlation between the high IF journals and high-citation papers and shorter length (everything, title, abstract, text). In this digital age, people do not have the time to navigate long texts with unwieldy language. Shorten and strengthen, for your own benefit.

Thanks very much for the feedback. We have reviewed and revised the entire manuscript making edits to phrasing and clarity where we thought possible. We have only highlighted major changes to specific sections, but multiple passes over the document have been made.

Reviewer 3 Report

Overall ok but a little addition on following observations will add value to the manuscript.

1.  Evaluated of various methods of selecting events in each batch that forms the input tensor to a camera rotation estimation network is claimed however it is not identified which one of these methods will lead to best/optimal selection.

2 The fact need to be established that Network training should be performed with the widest variety of batch sizes and velocities to produce a well generalised network.

Author Response

The authors appreciate and would like to thank the reviewer for the time and effort put in. We believe the manuscript improved based on the recommendations.

Evaluated of various methods of selecting events in each batch that forms the input tensor to a camera rotation estimation network is claimed however it is not identified which one of these methods will lead to best/optimal selection

We have strengthened our conclusions in section 5 to better clarify which would lead to better performance based on our analysis:
"The fixed-time batch is recommended for inference in a motion estimation network"
and Section 4.3:
"Therefore a simple fixed-time batching method is recommended for motion estimation network applications."

The fact need to be established that Network training should be performed with the widest variety of batch sizes and velocities to produce a well generalised network.

We highlight the first point of our conclusion section: 
"Network training should be performed with the widest variety of batch sizes and velocities to produce a well generalised network."
we believe the wording and manner of presentation should clearly communicate this conclusion.